# Usage of Terms "Science" and "Scientific Knowledge" in Nature of Science (NOS): Do Their Lexicons in Different Accounts Indicate Shared Conceptions?

**Ismo T. Koponen** 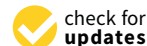

Department of Physics, University of Helsinki, 00014 Helsinki, Finland; ismo.koponen@helsinki.fi;
Tel.: +35-829-415-0652

**Abstract:** Nature of science (NOS) has been a central theme in science education and research on it for nearly three decades, but there is still debate on its proper focus and underpinnings. The focal points of these debates revolve around different ways of understanding the terms "science" and "scientific knowledge". It is suggested here that the lack of agreement is at least partially related to and reflected as a lack of common vocabulary and terminology that would provide a shared basis for finding consensus. Consequently, the present study seeks motivation from the notions of centrality of lexicons in recognizing the identity of disciplinary communities and different schools of thought within NOS. Here, by using a network approach, we investigate how lexicons used by different authors to discuss NOS are confluent or divergent. The lexicons used in these texts are investigated on the basis of a network analysis. The results of the analysis reveal clear differences in the lexicons that are partially related to differences in views, as evident from the debates surrounding the consensus NOS. The most divergent views are related to epistemology, while regarding the practices and social embeddedness of science the lexicons overlap significantly. This suggests that, in consensus NOS, one can find much basis for converging views, with common understanding, where constructive communication may be possible. The basic vocabulary, in the form of a lexicon, can reveal much about the different stances and the differences and similarities between various disciplinary schools. The advantage of such an approach is its neutrality and how it keeps a distance from preferred epistemological positions and views of nature of knowledge.

**Keywords:** nature of science; science education; networks; lexicons

## 1. Introduction

The characteristic features of science and scientific knowledge as they should be addressed in science education are topics that have been discussed for decades within science education and the research concerning it. In science education research, the specific research field focusing on this topic is known as Nature of Science (NOS) [1–6]. From the beginning of the NOS movement (see, e.g., Reference [1]), the focus of NOS and how it should be addressed has been debated in the research literature [7–9]. Some researchers emphasize the importance of broad characterisations of science, equally valid for all sciences [2,3,9], while some others have argued for discipline-specific approaches [10–14]. Opinions also differ in regard to the emphasis on the subjectivity as opposed to objectivity of scientific knowledge, as well as the nature of scientific knowledge as a construction that organizes experience, as opposed to the view that it describes reality and the real natural world as it is [15–19]. For over three decades, such differing positions have recurrently waxed and waned, and new viewpoints have been offered.

In science education and for practical teaching solutions, the most widely adopted viewpoint on NOS appears to be based on suggestions initially proposed by Lederman, McClough, Abd-el-Khalick, and their

collaborators (see, e.g., Reference [3,5,6]), consolidated by seven tenets (see e.g., Reference [3,20,21]): (1) the empirical nature of science, (2) the character of scientific theories and laws, (3) the creative and imaginative nature of scientific knowledge, (4) the theory-laden nature of scientific knowledge, (5) the social and cultural embeddedness of scientific knowledge, (6) the myth of (a single) scientific method, and (7) the tentative nature of scientific knowledge. These seven tenets form a backbone of the so-called consensus view on NOS [3]. The set of seven tenets and consensus view of NOS in science education have raised much criticisms (see, e.g., Reference [12,15,18,22,23]). A considerable part of the criticism reflect the positions of critiques against the philosophical underpinnings of NOS raised as early as three decades ago (see Reference [7,8]). The critics seem convinced that the consensus views of NOS needs to be augmented, if not abandoned and replaced altogether by some more well-founded and better-justified view (see, e.g., Reference [16,24]). The justifications of the alternative views are usually sought from science philosophy, in some cases blending several varieties of realistic positions and semantic views on theory (see, e.g., Reference [15,16,18,22,23]), while some authors prefer more un-blended views, like critical realism [19] (see also Reference [17,24,25]). One concrete alternative for consensus NOS in practical science education is the Family Resemblance Approach (FRA), suggested by Erduran and Dagher [26,27], based on the views of Irzik and Nola [22,23]. The FRA takes into account the disciplinary variation within sciences but recognizes that different scientific disciplines always have some sets of shared features; there is a family resemblance between and among disciplines.

The debates and discussions about NOS, its underpinnings and focal points and how they are related to "science", "scientific knowledge" and "scientific inquiry" appear to revolve mostly around the different ways to understand such key terms, which may have different meanings within different schools of thought [25]. This is a classic example of different paradigms, which lack the vocabulary and terminology that would provide a shared basis for finding a consensus [28–30]. The language and linguistic structures in communicating scientific results, ideas, and views is a central theme in Thomas Kuhn's conception of science, which he developed later in his studies [30]. This shift in Kuhn's philosophy of science has been referred as "Kuhn's Linguistic Turn" [28,29]. The linguistic turn consisted of paying attention to the relational structures of terms and concepts, and how they are interlinked and support the meaning of other terms and concepts through that networked, interlinked structure; such networks are made of clusters of interrelated terms. Accordingly, Kuhn refers to such networks as lexicons or lexical networks [28–30]. Each user of the lexicon has an option to use different interrelationships within the adopted lexicon, but to use that lexicon within a community, for communication, the concepts of the lexicon need to be used and conceived similarly enough by the members of the community. Kuhn refers to such a commonly shared understanding and use of lexicons as the structure of the lexicon [30]. The lexicons and structure of lexicons are, for Kuhn, the basic structure that allows us to recognize the disciplinary identity of a disciplinary group, and how the group frames and conceptualizes the basic phenomena of interest. Changes in lexicons and their structure also directly reflect changes in scientific understanding; the lexicons and the structure of lexicons are nearly substitutes for the disciplinary identity of research groups.

The present study uses the notions of centrality of lexicons in recognizing the identity of disciplinary communities and different schools of thought. Here, we apply such an approach to investigate how lexicons used by different authors to discuss NOS are confluent or divergent. While several previous studies have already addressed NOS and debates surrounding it from viewpoints based on history and philosophy of science and sociology of science, such viewpoints derive their arguments from chosen philosophical stances; the controversies thus emerging often go back to those underpinnings. Therefore, it is of interest to approach the problem from the viewpoint of lexicons and to pay attention how uses of different terms are related to the context of their use. Such context or thematic dependence is in the core of polysemy of the terms and words and may reveal the disciplinary differences in the usage of terms, thus revealing different ways to frame and rationalize the phenomena of interest [28–30]. The positions examined here were chosen to be the mainstream notion of NOS by Lederman [3] and the criticism towards it with suggestions of alternative

positions by Galili [16]. The third position examined is a mixed one, containing for most parts the underpinnings of the Family Resemblance Approach (FRA) by Erduran and Dagher [26,27], behavior including notions by Matthews [18]. The lexicons used in these texts are approached on the basis of a network analysis, recently introduced to analyze lexicons in students' written reports [31,32]. The results of the analysis reveal that the lexicons are different, in fact the most divergent views related to epistemology, but, regarding the practices and social embeddedness of science, one can find much basis for converging views.

## 2. Methods and Materials

Here, we analyze the lexicons that different authors use when they discuss their views of NOS, and how the lexicons parallel or diverge. Recognizing where lexicons meet and where they diverge may help to find a common ground for communication, to avoid unfruitful disputes that originate from unexplicated background views and philosophical biases, and to avoid talking past each other. Given that many key notions of NOS are currently not agreed and the scholarly publications also document clear disagreement and fault lines in dialogue over key issues (see, e.g., Reference [15–19]), such neutral and language related exploration seems warranted.

As three examples, we have chosen texts from (A) Lederman [20,21] and (B) Galili [16]. The third text collates excerpts from Erduran and Dagher [26,27], Irzik and Nola [23], and Matthews [18]. The decision to focus on these authors derives from the fact that: (1) they all discuss the same items, and (2) they represent clear differences in views. The research questions posed are:

RQ1: What are the lexicons (vocabulary) of texts A, B, and C?
RQ2: How do the lexicons of A, B, and C differ and where do they agree?

### 2.1. Formation of Text Corpus

The texts analyzed here discuss the consensus NOS (text A) and possible ways to extend it (texts B and C). Text A is composed of excerpts from two publications by Lederman [20,21], which summarize the seven basic tenets of the so-called consensus view of NOS. Each of the seven tenets are discussed at some length in the references on which text corpus A is based. The pages in each reference included in the corpus are listed in Table 1. The parts of texts that do not discuss the tenets, or that comment on works not related to the tenets, are omitted. Text B is based on a publication by Galili [16], which comments on and augments consensus NOS. The text by Galili is also organized around the seven tenets (which Galili calls the Lederman seven), so text B is structurally organized around the same thematic contexts as text corpus A. Text corpus C is different from A and B in that it is not based on a single author's publication but instead is comprised of excerpts from three sources: Erduran and Dagher [27], Irzik and Nola [22,23], and, Matthews [18]. The excerpts (with the references and pages given in Table 1) discuss the so-called Family Resemblance Approach (FRA) as introduced by Erduran and Dagher [27], its underpinnings as proposed by Irzik and Nola [22,23], and features of science, as discussed by Matthews [18] to the extent that FRA refers to them. The texts forming corpus C are thus not strictly organized around the seven tenets as in the cases of corpora A and B, but as FRA itself, based on a two-fold view of science as a cognitive-epistemic system and asocial-institutional system. However, within this division, it is possible to discern similar dimensions as those contained in the seven tenets in the text that forms corpus A. Note that, in all cases, the parts of texts that contain historical notions or specific examples are excluded. In what follows, the contents of each corpus are described briefly.

**Table 1.** Formation of text corpora A, B, and C.

| | Text | Excerpts ([Ref.] pp) |
|---|---|---|
| A | Lederman 2002, 2007 | [20] pp. 500–501, 502; [21], pp. 833–835. |
| B | Galili 2019 | [16] pp. 508–512, 513–514, 517, 518, 521, 523, 527, 528. |
| C | Erduran & Dagher 2019<br>Irzik & Nola 2011, 2014<br>Matthews 2012 | [27] pp. 312–314, 316 317.<br>[23] pp. 1004–1005, 1005–1006, 1014; [22] pp. 599–600<br>[18] pp. 9–12, 15, 16–18 |

Corpus A comprises key notions of consensus NOS. The consensus view of NOS originates from seminal views proposed in the late 1980s [1] and the basic tenets of that view are now consolidated in the form of seven notions [20,21]. Briefly, the seven tenets are as follows (for details, see, e.g., Reference [3,20,21]).

1. The empirical nature of science.
2. The character of scientific theories and laws.
3. The creative and imaginative nature of scientific knowledge.
4. The theory-laden nature of scientific knowledge.
5. The social and cultural embeddedness of scientific knowledge.
6. The myth of (a single) scientific method.
7. The tentative nature of scientific knowledge.

These seven tenets, each usually explained by a short description and examples (see, e.g., Reference [3,20,21]), form the core notions of the so-called consensus view of NOS. They consolidate, in a simple form, many ideas presented in sociologically oriented philosophy of science (items 3, 4, 5, and 7) and, in addition, notions developed in the so-called semantic view of science (items 1 and 2 and 4). Thus, it is easy to agree about the broad picture that these tenets paint about science. The consensus view also places attention on the difference between scientific inquiry and scientific knowledge, although such a distinction is more a practical choice for clarity than a fundamental difference.

Corpus B comprises key notions of Galili's criticism of consensus NOS [16]. In his critical account of the seven tenets, Galili suggests modifications for each item in the list of seven consensus items. In some cases, for example, in questions related to the tentative nature or the subjectivity of scientific knowledge, the suggested modifications take a strong realist stance and reject the constructivist tone of the consensus view. The views of Galili differ from the consensus view most in questions related to the epistemology of knowledge, subjectivity, and tentativity of scientific knowledge, and on the views of methods and methodology of science. In addition, Galili discusses at length the rationality of science, its theoretical structure and the role of idealizations and abstractions, as well as the mathematization of exact sciences. In general, Galili's focus is mostly on physics, while the consensus view attempts to embrace sciences more widely. In his critical account, Galili acknowledges how the list of seven items of the consensus view focuses on broad and general features of science and scientific knowledge, viable for discussions at the level of school. However, Galili claims that, in order to provide a deeper and more truthful picture, the tenets must be expanded significantly, and they need to be more sensitive to disciplinary differences.

Corpus C comprises key notions of the Family Resemblance Approach (FRA) by Erduran and Dagher [26,27] and its underpinnings as based on works by Irzik and Nola [22,23] and in some places making parallels with Matthews [18]. For these reasons, corpus C was constructed to contain not only descriptions of the FRA as found in refs. [26,27], behavior those descriptions the FRA borrows from philosophical analyses by Irzik and Nola and Matthews [18]. In that, corpus C differs from A and B, which are based on single author works. Following Irzik and Nola, the FRA sees science from two main perspectives: as a cognitive-epistemic system and as a social-institutional system. The part that is related to science as a social-epistemic system shares many similarities with consensus NOS in that it emphasizes the cultural and social embeddedness of science. The FRA is, however,

more explicit than consensus NOS about what it means to be "socially embedded" (e.g., institutions, organization, and means of communication) and it pays closer attention to professional activities and norms, the social certification of knowledge, and institutional factors. The part that relates to science as a cognitive-epistemic system discusses general aspects of behavior disciplinary differences in the process of scientific inquiry, its methods, and research approaches. The characteristic feature of the FRA is its sensitivity to disciplinary differences. Another specific feature of the FRA is its adherence to the conception that the core knowledge structures of science are theories, laws, and models and that these should be treated as related but different structures. The basic idea of the FRA is to recognize disciplinary differences, which apparently appear mostly in the cognitive-epistemic dimensions, but yet focus on those aspects that are shared by disciplines. Similar views focusing on features of science (FoS) and shared features, instead of some more consolidated nature of science common to all sciences, have been advocated by Matthews [18], who also speaks in favor of a more fine-grained and more discipline-sensitive understanding of science. In comparison to consensus NOS and FRA, however, FoS focuses more on the cognitive-epistemological aspects but not so much on the social-institutional aspects of science.

*2.2. Construction of Lexicons*

The lexicons corresponding to the text corpora A, B, and C are constructed as networks of words and terms: in the form of nouns, verbs and adjectives. The basic idea is to construct a stratified network in which syntactic structure is taken into account. Then, the distance of key words of interest in such a network can be quantified and used as a basis to measure how words co-occur in the texts. Such an approach can be taken as an extension of the n-gram analysis of word co-occurrences, through moving windows used to measure the distance of the words. The method we introduce here refines such a method by taking into account the different syntactic levels, from the level of sentence up to the level of context. We recently introduced such a method for the analysis of students study reports [31,32], as well as for didactic texts in science education [33].

The text excerpts that form the corpus to be analyzed consist of about 2500 words in A, 3200 in B, and 2200 in C. The original division of texts into sentences and passages is preserved and coded, *S* denoting sentence and KK the context formed by a complete passage. The passages are divided into narrower cotexts that discuss a more focused topic within a context. The number of contexts varies from 7 to 9, while the number of cotexts varies from 15 to 18 (each context containing 2–4 cotexts). The sentences *S* are divided into clauses and subclauses. In constructing the lexicons, clauses and sub-clauses are treated similarly so that nouns (*n*), verbs (*v*), and adjectives (*a*) are extracted, and the clauses and sub-clauses are treated as sets of nouns, verbs, and adjectives.

The lexicons of interest here are related to the terms T = [science] and T = [scientific knowledge], where brackets indicate that we treat them as lexical expressions. The lexicon is formed by all those terms and words that are linked to T through the lexical network. In addition to the terms and words, certain important adjectives are also taken into account and structurally treated like nouns. In the lexicon, the words and terms that are connected to terms of interest T are also connected to each other, on a similar basis, as they are connected to T. The basic idea is to construct a lexicon as a lexical network, which is stratified so that different levels of stratification indicate the syntactic position of the words in the lexicon. Finally, however, instead of a complete lexicon, the goal is to form a condensed representation of it, in the form of a lexicon profile. The lexicon profiles are formed in four steps by:

1.  Constructing the stratified lexical network from clauses *v* to contexts *KK*.
2.  Finding the most relevant terms through counting walks in the network.
3.  Constructing the lexical proximity network on the basis of walks.
4.  Extracting the key terms which form the lexicon profile for key terms T.

The construction of stratified lexical networks is performed so that we can differentiate levels, from single clauses to the cotext of several clauses and finally up to context. To accomplish this,

the pruned text consisting of main clauses is transformed into a network in which nodes corresponding to relevant terms and words (nouns) $n$ are first connected to nodes representing a root verb $v$. The root verb nodes $v$ are next connected to nodes $V$, each representing a clause or sub-clause, which finally form a sentence $S$. The sentences are then connected to cotext K, which form a context KK. After making these connections, the contexts are connected to represent the whole text corpus. A schematic representation of the part of the network which consists of connections from a set of nouns $\{n\}$ up to cotext K and context KK is shown in Table 2 with length of walks L needed to reach the term T from a given noun $n$. Note that T is part of set $\{n\}$, but the link is reversed $v \rightarrow T$.

**Table 2.** The construction of lexical networks. The form of links are shown in column "Link". The length L of walk that connects a given noun $n$ to term T of interest is given in column "Walk". Note that the term T is not shown, but it would be a link $v \rightarrow T$.

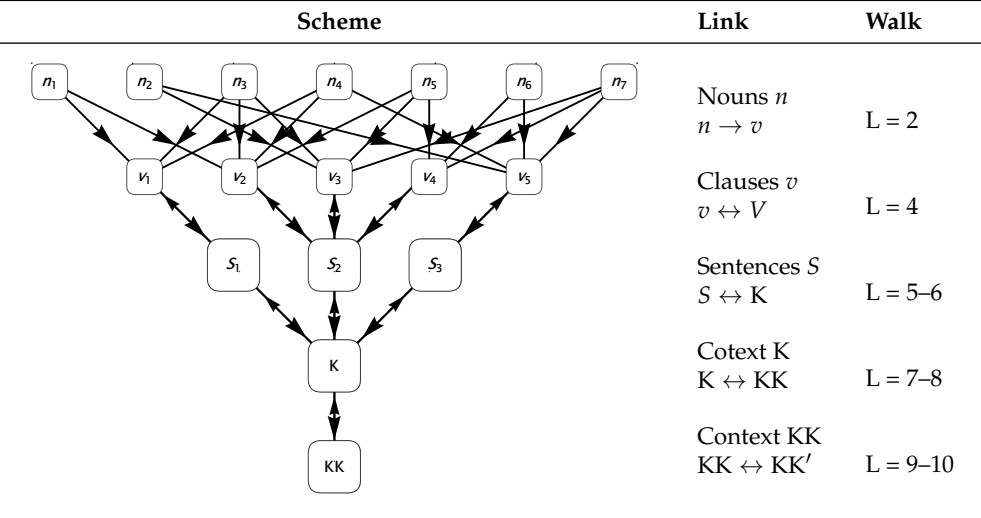

| Scheme | Link | Walk |
|---|---|---|
| | Nouns $n$ $n \rightarrow v$ | L = 2 |
| | Clauses $v$ $v \leftrightarrow V$ | L = 4 |
| | Sentences $S$ $S \leftrightarrow K$ | L = 5–6 |
| | Cotext K $K \leftrightarrow KK$ | L = 7–8 |
| | Context KK $KK \leftrightarrow KK'$ | L = 9–10 |

### 2.3. Analysis of the Lexical Network

The lexical networks, constructed from the level of clauses up to contexts, as explained in the figure in Table 2, are analyzed by paying attention to local and global connectivity. The lexical network can then be analyzed using network measures and metrics. The mathematical details of the measures used here are explained in Appendix A.

To summarize briefly, the most important measure we use is the so-called communicability $G_{pq}(\beta)$ between two nodes $p$ and $q$ in a network [34,35], which measures the connectivity of nodes in the network. The definition of communicability includes a parameter $\beta$ which allows weighting for local connections ($\beta \ll 1$) up to global connections ($\beta > 1$). The lexical distance of the words in the lexicon as measured through communicability thus reveals how a given word can be reached from another word, by traversing through the lexical network. The total communicability of a given node, its so-called communicability centrality $G_p = \sum_q G_{pq}$ [34–36], then describes the role of the node in connecting other nodes in the network as a whole. Recently, communicability and communicability centrality have shown to be useful and flexible quantifications for the analysis of lexical networks [31,32,37,38]. Here, we utilize the communicability in a slightly modified form, extracting the diagonal elements from the adjacency matrix. This amounts up to using so called matrix Laplacian instead of adjacency matrix [34,35] and helps to reveal even better the globally well-connected nodes. It should be noted that network-based analysis provides many alternative centrality measures, but, as we have rationalized elsewhere, the communicability is an obvious and well-rationalized choice to explore how semantic connections connect terms and words [31,32,37,38].

The network analysis based on communicability **G** provides, for each pair of nodes, the value $G_{pq}$, which can be interpreted as a measure of the proximity based on how likely the words corresponding

to the nodes appear to be related on some level of syntax (clause, sentence, cotext, and context); having more connections means higher communicability, which means higher proximity in the lexical network. In practice, the information about proximity can be now shown in form of a proximity network, where only pairs of nodes (words) that have a communicability exceeding a given threshold $G^*$ are connected. Such a pruned network of lexical proximity condenses the relevant information regarding how those words are connected that are of the greatest importance for the lexical meaning of other words.

### 2.4. Community Structure

The pruned proximity networks are also expected to show how certain words form dense clusters if they tend to appear in the same sentences or cotexts but are not so frequently encountered again in other cotexts that belong to other contexts. Such densely connected network structures, which can be recognized within the network, are called communities. Several methods exist for finding communities, each of them focused on different ways in which the communities are connected [39]. Here, we use the spectral method, suitable for networks with a so-called block structure, consisting of dense local clusters that are clearly separable from other local clusters. The spectral method is based on clustering nodes using the eigenvectors of the matrices. In performing the spectral clustering, the initial set of eigenvectors is transformed into a set of points with coordinates that correspond to the elements of the eigenvectors. The set of coordinate points is then clustered via standard k-means clustering [39]. Community detection using the spectral method is performed here using the `MATHEMATICA` function `FindGraphCommunities` [40].

### 2.5. Construction of Lexicons and Lexicon Profiles

The terms in the lexical proximity network are the basis for constructing, first, the lexicon and, second, the lexicon profiles based on those lexicons. The lexicons are the lists of key words and terms, recognized on the basis of their high communicability and content that is specific to the topic of interest (some common terms may have high communicability but are not of interest for the topic). The lexicon and lexicon profile are specific for the term T of interest (here T = [SCIENCE] or T = [SCIENTIFIC KNOWLEDGE]) but may of course overlap significantly. To form the lexicons and lexicon profiles, the words that are connected to T in the proximity network are classified among N categories, which condense the information of the lexical connections. Each category of the key terms describes a given descriptive property P = 1, 2 ..., N of interest. This classification is made for the practical purpose of condensing the relevant lexical information because lexical networks with complete interrelationships between terms are too rich to easily yield the relevant information regarding the lexical structure. The condensed representation of the lexicon in the form of a descriptive property of the words in a given category P, and with information on the relative importance of that category, is referred to as a lexicon profile.

The lexicon profile is formed by defining the lexical support $\Pi(P)$ the term T receives from the lexical proximity network of key terms $p(P)$ attached to a given feature P. Such support is operationalized as the sum of communicabilities $G_{Tq}$ of term $T$ to those words and terms $q$ that are contained in lexical proximity network, in the form

$$\Pi_P(T) = \sum_{p \in P} G_{Tp}. \tag{1}$$

The components $\Pi_P$ form the lexical profile as an N-dimensional vector of lexical supports $\bar{\Pi}(T) = (\Pi_1, \Pi_2, \ldots, \Pi_N)/\text{Max}[\Pi_P]$ where the normalization makes the lexicon profiles comparable.

## 3. Results

We first show the lexical networks for texts A, B, and C and then extract the lexicons in the form of lexical proximity networks containing the most important terms of the lexical networks. We also inspect the community structure of the networks. In addition, we introduce the 13 property classes for the construction of the lexicon profiles. The classification of words and terms on property classes is done on the basis of how related words refer to related targets. This classification is based on interpretation; thus, it is not as unambiguous as the division into communities, which is based purely on network metrics. Here, for terms T = [SCIENCE] and T = [SCIENTIFIC KNOWLEDGE], we introduce 13 property classes, as follows:

1. (SKW) Scientific knowledge in general and character of SKW.
2. (NOS) Nature of science or scientific knowledge, explicit reference.
3. (SCI) Science in general and character of science.
4. (THR) Theoretical knowledge and knowledge structures (e.g., models and laws).
5. (OBS) Observations, observers, and perceptions (excluding laboratory experiments).
6. (SIQ) Scientific inquiry, scientists and their activities, research activities.
7. (INF) Inference and explanation.
8. (TLD) Theory-ladenness (in a specific sense) of science and scientific knowledge.
9. (SUB) Subjective aspects of science, including imagination and creativity.
10. (SOC) Social and cultural embeddedness of science.
11. (RAT) Rationality, evidence, and truth in science.
12. (MTD) Scientific methods and methodological aspects.
13. (NAT) Nature and reality, natural and real phenomena.

In each lexicon, different sets of words and terms may appear in the property classes. The words that belong to each property class in the vocabulary of a given text corpora are decided by the authors; thus, the classification contains a component of interpretation. The division of words and terms into communities partially overlap with the division of words into property classes. These two divisions are complementary in the sense that the division into communities is based purely on text structure, reflecting the syntactic and context structure, while the division into property classes is based on an understanding of thematic similarities between words and terms. For each text corpus, the same set of 13 property classes is used in constructing the lexicon profiles.

### 3.1. Lexicon of Text Corpus A

The lexical networks corresponding to the texts A, B, and C are shown in Figures 1–3, respectively. The lexical network for text A shown in Figure 1 reveals a clear modular structure, consisting of about 11 communities in lexical networks for T = [SCIENCE] and T = [SCIENTIFIC KNOWLEDGE]. Both networks are very similar, with only small differences in the content of communities and how communities are interlinked. The lexical networks are shown for the local connectivity (local), corresponding to $\beta = 0.2$, and for global connectivity, with $\beta = 2.5$, which is large enough to retain the network unchanged when $\beta$ is increased. The most important words and terms in the communities are denoted by abbreviations, explained in Table 3. The size of the nodes in Figure 1 is proportional to the communicability centrality of the node, corresponding to the role of the node in supporting the lexical meaning of other nodes in the network; the larger the node, the more important it is in the lexical network. The values of the communicability centrality $G$ of the most important nodes shown in Figure 1 are given in Table 3. In Table 3, the communities of the words are provided.

From Table 3, we see that words or terms that refer to closely related targets, ideas, or properties are often in different communities. This, of course, reflects the fact that related topics are discussed in different contexts and in different ways. The communities thus emerge from contextualization of discussion, not from connections between words or terms on the basis of how they refer to related

targets (e.g., ideas or properties). In addition to the communities, words and terms can be classified into a property class $P_k$, with the property class given in Table 4. From Tables 3 and 4, we see that the terms and words that have the highest values of communicability provide the key terms for the communities. In the case of these most central words and terms, the division of these words into communities and property classes is to a large degree commensurate, each community roughly corresponding to one of the property classes. On the level of words and terms of intermediate and low values of communicability centrality ($G < 0.6$), the division into communities and property classes becomes less aligned. This can be interpreted so that certain key words, like "science" and "scientific knowledge", which thematically define the property classes, are also the words that thematically define the context of a text, but, in that context, auxiliary words that thematically belong to another context may appear. For example, in the context of scientific inquiry, "creativity" and "imagination" may appear, though these words characterize subjective human properties.

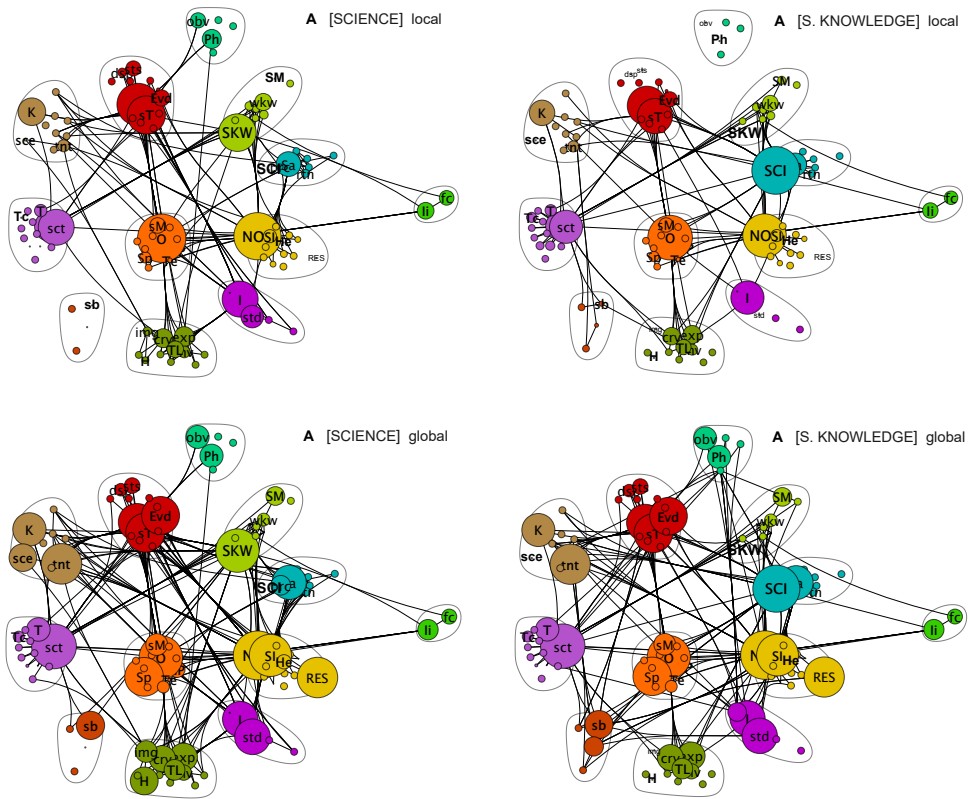

**Figure 1.** The lexical proximity network corresponding text corpus A. The lexical proximity networks for [SCIENCE] and [SCIENTIFIC KNOWLEDGE] are shown for threshold $G^* = 0.25$ corresponding to local (upper row) and global (lower row), obtained with $\beta = 0.2 \ll 1$ and $\beta = 2.5$, respectively. The size of the nodes corresponds to the communicability centrality $G_p$ (i.e., total communicability with all their nodes) of the node $p$. The communities of different colors are referred to using the most prominent common nodes (the node used as tag in tables is bolded) in them: Yellow (**NOS** & SI & He); Turquoise (**SCI** & Sa); Olive Green, light (**SKW** & SM); Orange (**O** & Sp); Red (**sT** & Evd); Brown (**K** & tnt); Purple, light (**sct** & T); Olive Green, dark (**TL** & H); Purple, dark (**I** & std); Red, dark (**Sb**); Green, light (**Ii**); Green, dark (**Ph** & obv). The acronyms of the symbols are explained in Table 3.

By comparing lexical networks that emphasize local connections (local, $\beta = 0.2$) to networks that place more weight on global connections (global, $\beta = 2.5$), we see that the relative centralities of the words and terms change slightly; in the global picture, scientific inquiry, human aspects, subjectivity

and theory-ladenness gain more weight than on the local scale. This suggests that text A discusses such topics in a contingent manner, picking up these themes in more than a few contexts. This indicates a multifaceted discussion of such topics. Of course, this conclusion has most probably been drawn also by simply reading the texts that form corpus A, but the advantage of networks analysis as shown in the figure in Table 2 is that it is not based on human interpretation (always subjective) but only on structural analysis that is independent of the content (and thus more objective).

**Table 3.** Lexicon of text corpus A. The words and terms for [SCIENCE] and [SCIENTIFIC KNOWLEDGE] with the highest communicability centralities $G_S$ and $G_K$ in text corpus A. The community (#c) and property class P of each word is indicated. The entries in boldface appear once each, either in [SCIENCE] or [SCI. KNOWLEDGE]. The abbreviations (abbr) and community (#c) abbreviations refer to Figure 3. The abbreviations for property classes P are explained in Table 4.

| Word or Term | Abbr | #c | P | $G_S$ | $G_K$ | Word or Term | Abbr | #c | P | $G_S$ | $G_K$ |
|---|---|---|---|---|---|---|---|---|---|---|---|
| NOS | NOS | NOS | NOS | 1.00 | 1.00 | Theory laden | TL | TL | TLD | 0.43 | 0.43 |
| Science | SCI | SCI | SCI | 0.97 | 0.97 | Inferential | Ii | Ii | INF | 0.42 | 0.42 |
| Scientist | sct | sct | SIQ | 0.92 | 0.92 | Perceptions | prc | SCI | OBS | 0.40 | 0.40 |
| Observation | O | O | O | 0.89 | 0.89 | Sci. method | SM | O | MTD | 0.38 | 0.38 |
| Sci. knowld. | SKW | SKW | SKW | 0.89 | 0.89 | Invention | inv | TL | SIQ | 0.32 | 0.32 |
| Sci. inquiry | SI | NOS | SIQ | 0.83 | 0.83 | Training | trn | sct | - | 0.26 | 0.26 |
| Tentative | tnt | K | SUB | 0.83 | 0.83 | Expectations | - | sct | - | 0.26 | 0.26 |
| Research | RES | NOS | SIQ | 0.83 | 0.83 | Way Knowng | wkw | SKW | SUB | 0.26 | 0.26 |
| Sci. theory | sT | sT | SKW | 0.80 | 0.80 | Descriptive | dsp | sT | - | 0.26 | 0.26 |
| Sci. process | sP | O | SCI | 0.79 | 0.79 | Thr. comts. | Tc | sct | TLD | 0.23 | 0.23 |
| Evidence | evd | sT | RTN | 0.78 | 0.78 | Preversus knowl. | - | sb | SUB | 0.20 | 0.20 |
| Sci. law | sL | sT | THR | 0.76 | 0.76 | Rational | rtn | SCI | RAT | 0.16 | 0.16 |
| Student | std | I | - | 0.73 | 0.73 | Legitimate | | sT | RAT | 0.15 | 0.15 |
| Inference | I | I | INF | 0.72 | 0.72 | Thr. entits | Te | sT | THR | 0.06 | 0.06 |
| Sci. activities | Sa | K | SIQ | 0.70 | 0.70 | Prior knwld | | sct | - | 0.06 | 0.06 |
| Knowledge | K | K | SKW | 0.68 | 0.68 | Char. knwld | chr | NOS | NOS | 0.03 | 0.03 |
| Subjective | sb | sb | SUB | 0.58 | 0.58 | | | | | | |
| Explanation | exp | TL | NOS | 0.58 | 0.57 | **Human** | H | TL | SUB | 0.57 | - |
| Creativity | crv | TL | NOS | 0.53 | 0.53 | **Soc. embedd** | sce | K | SOC | 0.56 | - |
| Belief | B | sct | SUB | 0.52 | 0.52 | **Imagination** | img | TL | SUB | 0.44 | - |
| Theoretical | T | sct | TLD | 0.48 | 0.48 | **Natural phen.** | nP | O | NAT | 0.33 | - |
| Phenomena | Ph | Ph | NAT | 0.47 | 0.47 | **Human entr.** | He | NOS | SUB | 0.29 | |
| Observable | obv | Ph | OBS | 0.47 | 0.47 | **Nat. wrld.** | - | sb | NAT | - | 0.38 |
| Sci. model | sM | O | THR | 0.45 | 0.45 | **Char. sci.** | – | I | SCI | - | 0.38 |

**Table 4.** The words and terms included in the nine property classes in text corpus A. The abbreviations of property classes are given under ABR.

| Property | ABR | Words and Terms Included |
|---|---|---|
| 1. Sci. knowld. | SKW | Scientific knowl., knowledge, body of knowledge., epistemolog(ical)/ly |
| 2. NOS | NOS | NOS, nature of science (NOS), characteristics of knowledge |
| 3. Science | SCI | science, characteristics of science |
| 4. Theory | THR | scientific theory, scientific law, scientific model, theoretical model, scientific concept, scientific entity |
| 5. Observation | OBS | observation, observable, observed, observer, perceptions |
| 6. Sci. inquiry | SIQ | scientific inquiry, inquiry, scientific activity, scientific investigation, scientist, invention, investigation, research, research program, technology, practitioner |
| 7. Inference | INF | inference, inferential, explanation |
| 8. Theory laden | TLD | theory laden, theoretical, theoretical entity., theoretical commitment, theoretical perspective |
| 9. Subjectivity | SUB | subjectivity, subjective, tentative, beliefs, mindset, way of knowing biases, construct, creativity, imagination,previous knowledge |
| 10. Soc. embedd. | SOC | socially embedded, culturally embedded., culture, human, human enterprise, values, politics, disciplinary commitment, power structure, social fabric |
| 11. Rational | RTN | rational, evidence, fact, certain, true, legitimate, foundational |
| 12. Method | MTD | scientific method, empirical, scientific process, data |
| 13. Nature | NAT | natural phenomenon, natural world, phenomenon, reality |

The comparison of lexical networks for T = [SCIENCE] to T = [SCIENTIFIC KNOWLEDGE] shows substantial similarity, although with some notable differences. For example, the words that refer to empirical aspects of science are weakly represented in [SCIENCE], while they are strongly represented in [SCIENTIFIC KNOWLEDGE]. This suggests that text A discusses the role of empirical results in relation to scientific knowledge but not so much in relation to science in general. Similarly, words referring to humans and constructions (see Table 3) are more strongly connected as part of the network of [SCIENTIFIC KNOWLEDGE] than as part of [SCIENCE]. This, of course, does not mean that such words should be understood as characterizing [SCIENTIFIC KNOWLEDGE], but they are discussed in the context of [SCIENTIFIC KNOWLEDGE] more than in the context of [SCIENCE]. Interestingly, the community containing words referring to reality, theoretical models, and inferential aspects of science became detached in the lexical network of [SCIENTIFIC KNOWLEDGE].

The lexical network contains a wealth of information regarding the lexical and syntactic structure of the text; thus, it is difficult to gain a consolidated picture of how central the given words are and how many of them are related; in other words, the thematic dimensions of the lexicon. In addition, the community structure, while providing important information on the content of contexts, remains relatively silent about the thematic dimensions and extension of the lexicon. This information is provided by the property classes and their 13 dimensions. Table 4 lists the words that are taken as key expressions in regard to the 13 thematic dimensions. The role and importance of each dimension as quantified in terms of lexicon profile $\Pi$ is based on the position of the words and terms in the lexical network, thus implicitly also containing the information of the community structure, through the values of the communicabilities of words contained in the lexicon.

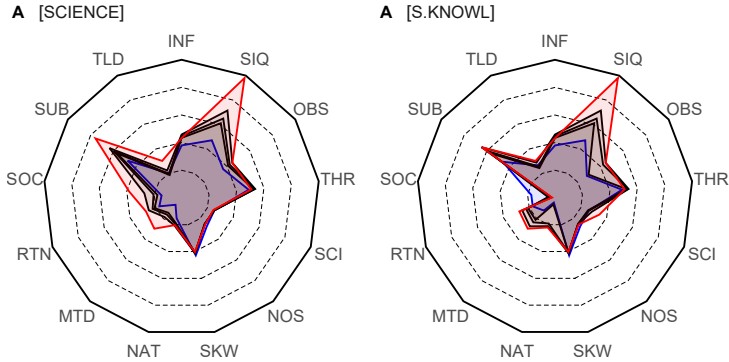

**Figure 2.** The lexicon profiles $\Pi$ proximity network corresponding to text corpus A. The abbreviations referring to lexical dimensions are explained in the main text and the words contained in each dimension in Table 4. The results are shown for local ($\beta = 0.2$, blue line) and for global connections ($\beta = 2.5$, red line), while the intermediate cases with $\beta = 0.5, 1.0$ and $1.5$ are shown in black lines.

The 13-dimensional lexicon profiles for text A shown in Figure 2 reveals how, with increasing syntactic depth, from the level of clauses to the level of connected contexts, certain dimensions become strengthened. With increasing values from $\beta = 0.2$ up to $\beta = 2.5$, the analysis gradually covers levels of L from 1 and 5 up to levels L ≤ 7, where more remote connections start to contribute. These more remote connections increasingly provide the semantic, context-related connections to the lexical terms. The more remote connections are supposedly important in providing the semantic content and also the different contextual ways to understand the meaning of terms, i.e., they reveal the context-relatedness and dependence of lexicons.

For T = [SCIENCE], the dimensions of subjectivity of knowledge (SUB), scientific inquiry (SIQ), theory (THR), and scientific knowledge (SKW) are nearly equally weighted on the local level (clauses and sentences), but, with increasing syntactic depth when the context level is taken into account, SUB and SIQ dominate the other dimensions. However, the dimensions of social embeddedness (SOC), rationality (RTN), and methodology (MTD) also gain importance but retain a relatively low

significance. Obviously, the themes related to SUB and SIQ are discussed throughout the text so that their lexical support (i.e., how they attach to other words) becomes enriched; new meanings become attached through new words. For T = [SCIENTIFIC KNOWLEDGE], the thematic dimensions are very similar to T = [SCIENCE], but now the role of scientific inquiry (SIQ) is even more dominant. The role of social embeddedness is diminished to insignificance, indicating that it is not discussed much in the context of scientific knowledge. The close similarity (apart the role of SOC) is perhaps to be expected because, in text A, it is mentioned that nature of science mostly concerns aspects of nature of scientific knowledge, thus collating many aspects of science with aspects of scientific knowledge. The curious feature of the lexicon profiles, however, is the strong emphasis on vocabulary related to subjectivity (SUB) of knowledge and relatively low significance of vocabulary related to rationality (RTN) and natural phenomena, including reality (NAT). This is perhaps interpretable as a sign of the strong constructivist stance of text A, which is also reflected in the vocabulary of that text.

Finally, when the results for the thematic 13 dimensions are compared with the community structure, we can see that they correspond to each other reasonably well, although with certain clear differences. This can be taken as an indication of coherence in thematic and contextual structures, which obviously results from the fact that text A is arranged around the seven tenets, which organize the text and provide it with an overall systematic structure.

### 3.2. Lexicons of Text Corpus B

The detailed analysis of the lexical networks and lexicon of text A shows that the most interesting and important information about the lexicons can be consolidated in the form of a lexicon profile. The community structure is interesting, but dependent on context, and furthermore, the key words and terms that are central to the lexicon are essentially the same words and terms that act as key-terms in the property classes. Therefore, in what follows, we focus mostly on lexicons. The pruned lexical networks for text B are shown in Figure 3, a summary of key terms and words with their community and property class associations in Table 5, and the list of key terms included in the definition of lexicon profiles in Table 6. In Table 5, the  abbreviations used in the figures are given, which allows us to read off the community structures of the networks.

The lexical networks corresponding to text B have a community structure resembling that of text A, reflecting a structure formed around the seven tenets. This is of course expected because text B is organized around the seven items in the consensus NOS list; thus, the contexts of discussions can be expected to be similarly structured in texts A and B. That the structures of the lexical networks are rather similar shows that the network captures the structural features inherent in the text corpora. When, however, we take a closer look at the words that belong to the communities, we find that a somewhat different wording is apparent in each text corpora B in comparison to A. We also see how words that belong to different property classes, formed on the basis of the mutual resemblance of the words and the targets they refer to, are distributed over different communities. This means that the same topics (related to words in similar property classes) are discussed in different contexts (related to communities).

In the case of text B, the lexical networks for [SCIENCE] are better connected than the networks for [SCIENTIFIC KNOWLEDGE]. In both cases, however, the communities including words and terms related to theory (THR), scientific knowledge (SKW), and reality and certainty (R) play central roles. Interestingly, the community corresponding to NOS becomes weakly connected in the network corresponding to [SCIENTIFIC KNOWLEDGE], revealing that, in text B, NOS is not much connected to discussions concerning scientific knowledge, although it plays an important role in discussing science. Moreover, the vocabulary related to science (recall that the term [SCIENCE] is excluded from lexical network T = [SCIENCE], but other terms and words of property class P = Science in Table 6 are included) is weakly present in lexical network T = [SCIENCE] but central in T = [SCIENTIFIC KNOWLEDGE]. Similarly, the vocabulary related to the term [SCIENTIFIC KNOWLEDGE] is weakly present in the lexical network for T = [SCIENTIFIC KNOWLEDGE] but central to the network of

T = [SCIENCE]. This kind of asymmetry was absent in the case of text A, which addressed [SCIENCE] and [SCIENTIFIC KNOWLEDGE] with more symmetric vocabularies. The asymmetry in the case of text B indicates that it approaches science and scientific knowledge as different kinds of thematic topics and does not collate them to the same degree as text A.

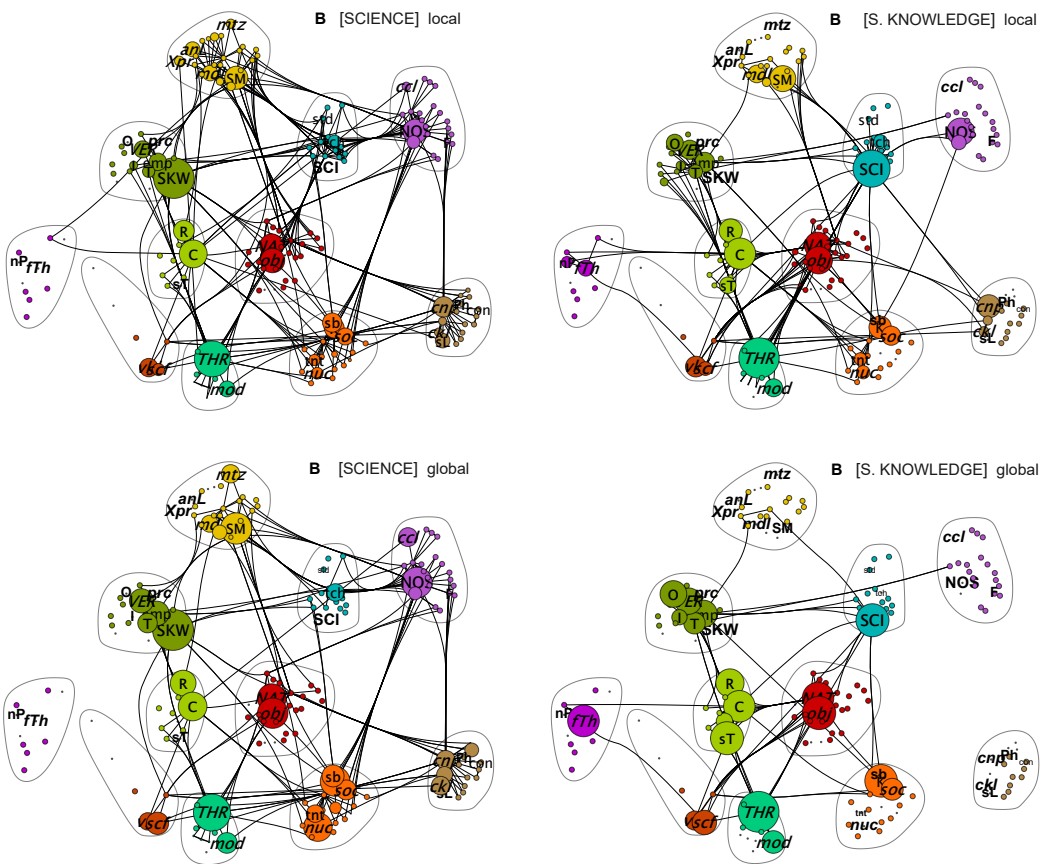

**Figure 3.** The lexical proximity network corresponding to text corpus B. The lexical proximity networks for [SCIENCE] and [SCIENTIFIC KNOWLEDGE] are shown for threshold $G^* = 0.35$ corresponding to local (upper row) and global (lower row) with $\beta = 0.2 \ll 1$ and $\beta = 2.5$, respectively. The size of nodes corresponds to the communicability centrality $G_p$ (i.e., total communicability with all their nodes) of the node $p$. The communities are shown with different colors, and the most important nodes in the communities are denoted by symbols explained in Table 5.

The lexical networks show that, for text B, the local ($\beta = 0.2$) are better connected than the global ($\beta = 2.5$) lexical networks, and, in the global networks, certain terms begin to dominate. This shows that, for only a few terms and words, lexical support becomes strengthened when longer connections are established at the level of cotexts and contexts are taken into account. Most of the terms and words appear only locally, on the level of sentences, but are not encountered again in sentences that belong to other contexts. This, on the other hand, means that a few important connections already introduced on the level of clauses and sentences are repeated or reiterated frequently; the text emphasizes the same aspects in many cotexts and contexts. In particular, many such terms and words emerging recurrently are related to methodology, rationality, and theory. This feature of the text is clearly visible in lexicon profiles.

**Table 5.** Lexicon of text corpus B. The words and terms for [SCIENCE] and [SCIENTIFIC KNOWLEDGE] with the highest communicability centralities $G_S$ and $G_K$. The abbreviations (abbr) and community (#c) are as in Figure 3, and property classes P are explained in Table 6. The items appearing only in one lexicon are in boldface.

| Word or Term | Abbr | #c | P | $G_S$ | $G_K$ | Word or Term | Abbr | #c | P | $G_S$ | $G_K$ |
|---|---|---|---|---|---|---|---|---|---|---|---|
| Sci. knowld. | SKW | SKW | SKW | 1.00 | | **Coll. knowl.** | ckl | cnp | SUB | 0.50 | - |
| Theory | THR | THR | THR | 0.91 | 0.91 | **Mathematiz.** | mtz | SM | MTD | 0.50 | - |
| **Sci. methd.** | SM | SM | MTD | 0.79 | - | **Tentative** | tnt | soc | SUB | 0.48 | - |
| **NOS** | NOS | NOS | NOS | 0.77 | - | Valid | vld | scf | RTN | 0.48 | 0.48 |
| Objective | obj | NAT | NAT | 0.76 | 0.76 | **Teaching** | tch | SCI | - | 0.48 | - |
| Science | SCI | O | OBS | | 0.76 | **Modeling** | mdl | SM | MTD | 0.47 | - |
| Knowledge | K | soc | SKW | 0.75 | 0.75 | **Cult. prod.** | - | cnp | SUB | 0.45 | - |
| Certain | C | R | RTN | 0.73 | 0.73 | Systems | - | SCI | RTN | 0.43 | 0.43 |
| Nature | NAT | NAT | NAT | 0.73 | 0.73 | **Explanation** | exp | cnp | INF | 0.40 | - |
| **Nucleus** | nuc | soc | | 0.73 | - | **Phenomena** | Ph | cnp | NAT | 0.39 | - |
| Reality | R | R | RTN | 0.66 | 0.79 | **Sci Educ** | - | NOS | | 0.38 | - |
| Empirical | Emp | SKW | MTD | 0.64 | 0.64 | **Accurate** | - | SM | MTD | 0.35 | - |
| Verification | VER | VER | RTN | 0.63 | 0.63 | **Fact** | F | NOS | RTN | 0.31 | - |
| Scientific | scf | scf | SKW | 0.59 | 0.59 | **Experimntl.** | Xpr | SM | MTD | 0.16 | - |
| Subjective | sb | soc | SUB | 0.72 | 0.72 | **Epistemol.** | - | SKW | SKW | 0.15 | - |
| Sci. activity | Sa | K | SIQ | 0.70 | 0.70 | **Systm. knwl.** | - | SKW | SKW | 0.13 | - |
| Knowledge | K | K | SKW | 0.68 | 0.68 | **Philosophy** | Ph | cnp | - | 0.12 | - |
| **Subjective** | sb | soc | SUB | 0.58 | - | | | | | | |
| Model | mod | THR | THR | 0.55 | 0.55 | **Sci. thr** | sT | R | SKW | - | 0.76 |
| Soc. env | soc | soc | SUB | 0.54 | 0.54 | **Fnd. thr** | fTh | fTh | SKW | - | 0.73 |
| Theoretical | T | SKW | THR | 0.53 | 0.53 | **Observation** | O | O | O | - | 0.63 |
| **Concept** | cnp | cnp | SKW | 0.52 | - | **Inference** | I | I | INF | - | 0.29 |
| **Conceptual** | ccl | NOS | SKW | 0.52 | - | **Coherence** | - | R | - | - | 0.27 |

**Table 6.** The words and terms included in the nine property classes (with abbreviations ABR) in text B. Repeated constructs are given as X+{...}. Expressions in parenthesis may appear without X.

| P | ABR | Words and Terms Included |
|---|---|---|
| 1. | SKW | Scientific knowledge, disciplinary knowledge, systemic knowledge., knowledge element, epistemolog(ical)/(ly) |
| 2. | NOS | NOS, nature of science (NOS) |
| 3. | SCI | science, structure of science, progress of science, scien(ce)/(tific) products |
| 4. | THR | scientific theory, scientific law, model(ing), scientific concept, structure of theory, conceptual foundation, conceptual system, conceptual structure, fundamental theory |
| 5. | OBS | observation, observable, observed |
| 6. | SIQ | scientific + {inquiry, activity, enterprise., investigation, discourse}, research, research + program, group, project, discovery, knowledge + {production, justification} scientist, researcher, practitioner |
| 7. | INF | inference, explanation, deduction, interpretation |
| 8. | TLD | theory laden, theoretical, theory, theoretical + {activity, product, perspective} |
| 9. | SUB | subjectivity, subjective, tentative, belief, tentative knowledge, creativity, imagination |
| 10. | SOC | community, cultural/social environment, cultural product, disciplinary culture, human knowledge, collective knowledge construction, social impact, values |
| 11. | RTN | rational, verification, objectiv(e)/ity, objective, certain, accura(te)/cy, truth factual, valid, logical + {product, proof, rule}, self-correcting, hypothetic-deductive |
| 12. | MTD | scientific method, empirical, method/(ology), methodological + {program, foundation}, experimental/(ity), mathematization, data, laboratory, technemethod, technology, operational, experimental activity, measurement, apparatus |
| 13. | NAT | nature, reality, phenomena, physical |

The lexicon profiles corresponding to the text B are shown in Figure 4 for T = [SCIENCE] and T = [SCIENTIFIC KNOWLEDGE]. As it has been already visible from lexical networks, text B has an extensive vocabulary related to rationality (RTN), methodology (MTD), and theory (THR), as well as theory-ladenness (TLD). In the lexicon profile for [SCIENTIFIC KNOWLEDGE], the rationality (RTN) and theory (THR) dimensions are very dominant and become more dominant over other dimensions when syntactic depth increases from the level of sentences up to the level of contexts. Consequently, rationality, theory, and methodology are apparently, on the basis of the vocabulary, topics that pertain to the whole text and are thus centrally important for the text. Such persistent emphasis on the topics of rationality, theory, and methodology is indeed evident from the general tone of text B, in that it criticizes consensus NOS on the lack of proper attention to just these topics and features of science and scientific knowledge. It is of course satisfying to find that the emphasis and general tone of the text, as apparent to a reader of the text, becomes so clearly visible in the simple (and interpretation-neutral) analysis of the vocabulary.

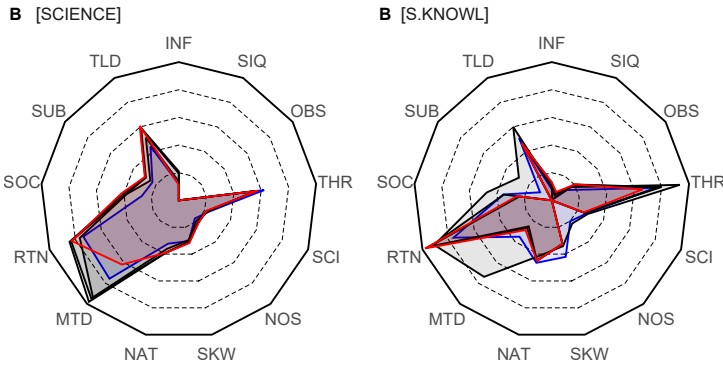

**Figure 4.** The lexicon profiles $\Pi$ proximity network corresponding to text corpus B. The abbreviations referring to lexical dimension are explained in the main text, the words contained in each dimension in Table 6. The results are shown for local ($\beta = 0.2$, blue line) and for global connections ($\beta = 2.5$, red line), while the intermediate cases with $\beta = 0.5, 1.0$ and $1.5$ are shown in black lines.

### 3.3. Lexicon of Text Corpus C

Text corpus C is aggregated from three different sources, mostly from texts related to the so-called Family Resemblance Approach (FRA) on NOS and a source explaining its philosophical underpinnings (see Table 1). In addition, the corpus includes notions extracted from the text explaining views called features of Science on NOS (see Table 1). The fact that the text is aggregated means that it contains a more varied and disparate set of words, is more fragmented, and the contexts are not as well aligned with the seven tenets as in corpora A and B. These features become visible in lexical networks as they are shown in Figure 5, whereas, on the local scale, we see a few dense communities (red, yellow, green, and purple), but, on the global scale, only the red and purple communities remain dense. This behavior reflects the fact that we can find richer connections on the level of sentences than on the level of contexts because the same words most often appear only in a limited number of contexts and are not encountered recurrently in other contexts; the text does not reiterate the ideas and views in several contexts, thus the meaning of words remaining as they are formed on the level of sentences or at most at the level of cotexts. Furthermore, that behavior is strengthened because text C contains extensive lists of words and terms to exemplify the topics discussed.

The lexical networks in Figure 5 for T = [SCIENCE] and T = [SCI. KNOWLEDGE] are somewhat different, the differences being greater than what was encountered in the networks for texts A and B. The most prominent terms listed in Table 7 show that now much fewer terms are shared, and [SCI KNOWLEDGE] features many terms not found in [SCIENCE]. Such behavior indicates that text C discusses science and scientific knowledge in a different way. Of course, this is also evident

from text C itself, since it makes the distinction between science as acognitive-epistemic system and a social-institutional system. The results in Figure 5 and Table 7 only quantify these differences through the metric of lexical distances of the words.

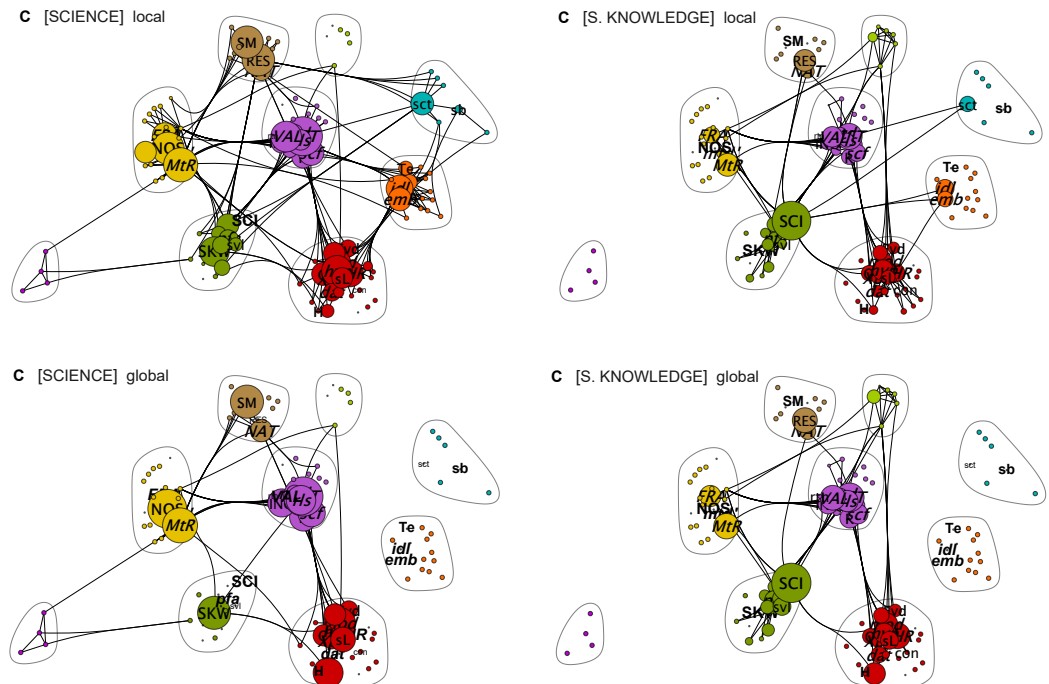

**Figure 5.** The lexical proximity network corresponding to text corpus C. The lexical proximity networks for [SCIENCE] and [SCIENTIFIC KNOWLEDGE] are shown for threshold $G^* = 0.25$ corresponding to local (upper row) and global (lower row), obtained with $\beta = 0.2 \ll 1$ and $\beta = 2.5$, respectively. The value $\beta = 2.5$ is high enough to guarantee that complete networks from level of clauses up to contexts are taken into account, when results remain intact with increasing value of $\beta$. The size of the nodes corresponds to the communicability centrality $G_p$ (i.e., total communicability with all their nodes) of the node $p$. The communities are shown with different colors, and the most important nodes in the communities are denoted by the symbols explained in Table 7.

In corpus C, the richness of vocabulary is nearly twice as extensive as in A, but most words and terms appear only once or twice. The text corpus has a character of listing words, with little connection to more in-depth discussions. This, of course, results from the way the corpus is constructed, from excerpts summarizing key notions contained in the text that provide the underpinnings of notions discussed elsewhere. In particular, the property class SOC, containing words that refer to social and organizational aspects of science, is extensive. In Table 7, only about 40% of words in that class are listed. However, the words that occur with low frequency (once or twice, mostly), and are thus not included, have a negligible effect on the lexicon profiles. In addition to the property class SOC, the property class of methods (MTD) also contains many words and terms that are mentioned only a few times and thus are omitted in Table 7, which thus contains only about 45–50% of all possibly relevant items. The community analysis collates the words that belong to property classes SCI (science) and SKW (scientific knowledge). Similarly, communities sL and VAL contain many words and terms that belong to different property classes. These aspects mean that the same words may appear in very different contexts of discussion and thus have different weightings of meaning. For example, the methodological rules can be discussed from the point of view of scientific methods but equally well from the point of view of the institutional norms of doing science.

**Table 7.** Lexicon of text corpus C. The words and terms for [SCIENCE] and [SCIENTIFIC KNOWLEDGE] with the highest communicability centralities $G_S$ and $G_K$, respectively, in text corpus C. The community (#c) and property class P of each word is indicated. The entries in boldface appear once, either in [SCIENCE] or [SCI. KNOWLEDGE]. The abbreviations (abbr) and community (#c) abbreviations refer to Figure 5. The abbreviations for property classes P are explained in Table 8.

| Word or Term | Abbr | #c | P | $G_S$ | $G_K$ | Word or Term | Abbr | #c | P | $G_S$ | $G_K$ |
|---|---|---|---|---|---|---|---|---|---|---|---|
| SCIENCE | SCI | SKW | SCI | | 1.00 | Experimentl. | Xpr | sL | MTD | 0.36 | 0.36 |
| **Method** | MET | VAL | MTD | 0.75 | - | Methodolg. | mtd | SCI | MTD | 0.35 | 0.35 |
| **NOS** | NOS | NOS | NOS | 0.68 | - | **Human** | H | sL | SOC | 0.32 | - |
| Meth. rules | MtR | NOS | MTD | 0.68 | 0.68 | **Theory** | THR | sL | THR | - | 0.68 |
| **Sci. meth.** | SM | SM | MTD | 0.64 | - | **Research** | RES | SM | SIQ | - | 0.66 |
| SCI. KNOWL. | SKW | SKW | SKW | | 0.64 | **Values** | VAL | VAL | SOC | - | 0.66 |
| Models | mod | sL | THR | 0.62 | 0.62 | **FRA** | FRA | NOS | NOS | - | 0.58 |
| Sci. entrpr. | sce | sL | SIQ | 0.59 | 0.59 | **Hypothesis** | hyp | sL | MTD | - | 0.55 |
| Rules | rLs | VAL | MTD | 0.58 | 0.58 | **Prof. actvts.** | pfa | SKW | SIQ | - | 0.53 |
| Inquiry | INQ | VAL | SIQ | 0.56 | 0.56 | **Soc. values** | svl | SKW | SOC | - | 0.50 |
| Knowledge | K | VAL | SKW | 0.55 | 0.55 | **Data** | dat | sL | MTD | - | 0.45 |
| Scientific | scf | VAL | SCI | 0.54 | 0.54 | Sci. ethos | sce | SKW | SOC | - | 0.44 |
| **Fact** | F | SCI | RTN | 0.52 | - | Sci. activt. | Sa | SKW | SIQ | - | 0.40 |
| Sci. law | sL | sL | THR | 0.50 | 0.50 | Soc. certf. | scr | SKW | SOC | - | 0.40 |
| Nature | NAT | SM | NAT | 0.45 | 0.45 | **Norms** | nrm | SCI | SOC | - | 0.37 |
| Explanation | exp | sL | INF | 0.45 | 0.45 | Soc. inst. | sin | SCI | SOC | - | 0.34 |
| Observation | O | sL | OBS | 0.44 | 0.44 | **Cog. Epistm.** | cge | VAL | SKW | - | 0.27 |
| Evidence | evd | sL | RTN | 0.38 | 0.38 | **Rational** | R | VAL | RTN | - | 0.18 |

**Table 8.** The words and terms included in the nine property classes (with abbreviations ABR) in text C. Repeated constructs are given as X + {…}. Expressions in parentheses may appear without X.

| P | ABR | Words and Terms Included |
|---|---|---|
| 1. | SKW | Scientific knowledge, knowledge, cognitive-epistemic, epistemic control |
| 2. | NOS | NOS, FRA, family resemblance, FoS |
| 3. | SCI | science, character of science, features of science, scientific, new science |
| 4. | THR | scientific + {theory, law, model(ing), concept } |
| 5. | OBS | observation, observable, observation method, observation report |
| 6. | SIQ | scientific +{enterprise, inquiry, activity, investigation, research}, scientists, research program, technology |
| 7. | INF | inference, inferential, (scientific) explanation, explanatory structure, interpretation |
| 8. | TLD | theory laden, theoretical, theoretical +{entity, construct, choice} |
| 9. | SUB | subjectivity, subjective, tentative, beliefs |
| 10. | SOC | socially embedded, social + {values, norm, activity, certification, institution} cultural, social, political, scientific + {community, ethos}, human |
| 11. | RTN | fact, rational, objectiv(e)/ity, evidence, truth, reliable, testable |
| 12. | MTD | (scientific) method, methodology, methodological rule, empirical, empirical basis data, measurement, experiment, measuring instrument, analytical method |
| 13. | NAT | nature, phenomenal reality |

The major part of the text corpus in C is about the FRA (Family Resemblance Approach), which makes a clear distinction between the cognitive-epistemic and the social-institutional aspects of science. These focal points of discussion are also reflected in the lexicon profile shown in Figure 6, where we see a strong focus on methodological aspects and scientific inquiry, as well as on social aspects. This division of topics into two broad categories might also explain why science and scientific knowledge overlap in the community analysis. Similarly, the overlap of models with factors related to observation, rationality, and partly also to norms may be explained by how FRA in the cognitive-epistemic part focuses on the roles of models, theories, and laws as structures of scientific knowledge. The fact that text C introduces topics contextualized in two broad thematic areas is visible from the formation of dense clusters, where words referring to different thematic areas,

like methodology, knowledge structure types, and even norms, become connected. On the other hand, some words related to those thematic areas on the local scale fall out of the dense clusters on the global scale, for example, the words: empirical basis (emb), theoretical entities (Te), and idealization (idl).

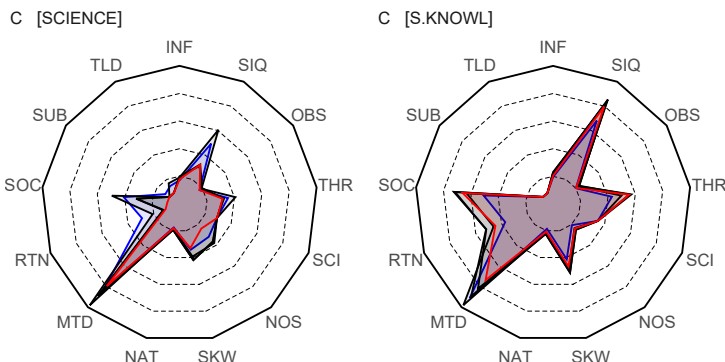

**Figure 6.** The lexicon profiles **Π** proximity network corresponding to text corpus C. The abbreviations referring to lexical dimensions are explained in the main text, the words contained in each dimension in Table 7. The results are shown for local ($\beta = 0.2$, blue line) and for global connections ($\beta = 2.5$, red line), while the intermediate cases with $\beta = 0.5, 1.0$ and $1.5$ are shown in black lines.

A closer look at how different terms become divided into different communities reveal their closeness when they become contextualized. The community analysis, while sensitive to contextualization, may establish relationships that are difficult to understand on the basis of the thematic closeness of the words. For example, words referring to theoretical entities may appear in communities that are different from those where models and modeling appear. The way the words appear in communities is more sensitive to contextualization than thematization, whereas, for different contextualizations (depending on viewpoints taken), more contingent choices are possible than for thematical categorizations.

### 3.4. Comparisons of Text Corpora A, B, and C

The lexicon profiles **Π** corresponding to text corpora A, B, and C can be used as bases for comparison that are based on thematic categorizations of words and are thus less sensitive to context than the community structure. As we have seen, in some cases (text A), the community structure parallels the thematic categorization, but these may also be essentially different (as in text C). This is one consequence of abstract words being polysemous; they do not have similarly fixed meanings as words referring to simple concrete objects. Therefore, in comparing the lexicons, we focus on lexicon profiles as based on thematic relatedness instead on community structure, which reflect context relatedness. The lexicon profiles, instead of being context dependent, attach thematically related (i.e., same kinds of referents or even synonymous) words and terms to [SCIENCE] and [SCIENTIFIC KNOWLEDGE]. This information on thematical dimensions is reduced to 13-dimensional vectors, where each dimension is denoted by one of the tags listed in Tables 4, 6, and 8 for property class P. The choice of key words in Tables 4, 6, and 8 is specific to the text, some of them identical, but generally different vocabularies are used. Therefore, the choice of key words contains an element of interpretation about the meaning of the word and to what it refers. Nevertheless, as is seen in the listings of key-words in Tables 4, 6, and 8, such classification contains fewer unexpected groupings of words than the grouping based on community structure.

The lexicon profiles corresponding to texts A, B, and C in Figures 2, 4, and 6 are collected for comparison in Figure 7. It is seen that text A covers nearly all aspects contained in the 13-dimensional profile, thus having a balanced weight of vocabularies to discuss all 13 aspects. Some of the vocabularies, like methodology (MTD), rationality (RTN), and nature/reality (NAT), are under-weighted in comparison to other dimensions. In addition, the similarity of lexicon profiles for T = [SCIENCE] and T = [SCI. KNOWLEDGE) is noteworthy. The similarity is in agreement with the notion contained in the

text that it sees NOS and nature of scientific knowledge (NOSK) as similar constructions. The lexicon profile puts much weight on items related to scientific inquiry (SIQ) and the subjectivity and tentativity of knowledge (SUB). These notions are also in line with the purpose stated in the text. It should be borne in mind that the results condensed in lexicon profiles are obtained purely by inspection of vocabulary and by measuring the lexical distance of words, without interpreting the content of the text (the only interpretative element is the choice of key-words, which is not guided by an interpretation of the text, but the general understanding of the authors who have backgrounds in physics and physics education).

The lexicon profile for text B is very different from that of A. Text B, as it appears, was written to extend and augment the views contained in text A and other similar texts. The dimensions MTD, RTN, and NAT, which were underrepresented in A, are now heavily overrepresented in B. In this, text B clearly serves the purpose its author mentions: to strengthen and augment the aspects omitted in consensus NOS. With regard to [SCI. KNOWLEDGE], text B has a very extensive vocabulary of items related to theory [THR], revealing a stronger emphasis on theoretical knowledge in comparison to text A. The differences between the vocabularies of A and B are so significant that it raises questions as to whether the two texts have any common ground at all. If we take at a face value the notion that lexicons and their differences indicate differences in thought and ways of framing the phenomena and posing the relevant questions, we can conclude that texts A and B indeed represent different schools of thought.

The lexicon profile for text C reveals an emphasis on methodology to the same degree as the lexicon profile for B, but now scientific inquiry and the sociological and institutional aspects also have strong vocabularies. Lexicon profile C clearly augments the aspects that are not so well represented in A, yet retains the certain richness of dimensions, although not to the same degree as A. The lexicon profile for C is not as far removed from A as B, but, in C, the dimensions related to scientific inquiry (SIQ) and methodology (MTD) are also over-emphasized in comparison to other dimensions. In general, the lexical network for C reveals a certain fragmentation and lack of coherence, but this is clearly a consequence of how the corpus of C was constructed as a collated corpus.

The lexicon profiles of A, B, and C are quite different, which can be interpreted in two ways. The first possible interpretation is that both B and C have grown from criticism towards consensus view of NOS (text A); thus, they pay attention to dimensions that their authors think text A misses or misrepresents. On reading the texts, the impression is indeed that this is clearly the goal of text B, but not so much of text C. The second interpretation is that texts B and C are meant to augment the views represented in text A, not so much to criticize and replace the consensus view. However, if this second interpretation is viable, one would expect the vocabularies of texts of B and C to overlap more with the vocabulary in A. From the perspective of overlap of vocabularies, as represented by the lexicon profiles, the common ground for easy and co-constructive discussion is shallower than might be desired. Therefore, the first interpretation, that texts B and C can be taken rather as criticism and attempts to suggest alternative views, gains credibility if one focuses only on vocabulary and how it overlaps with A. Such a visible discrepancy of vocabularies may well be one reason behind the slow convergence of views and perhaps the even formation of different schools of thought concerning NOS. However, from the viewpoint of lexicons and vocabulary, the consensus view as consolidated in seven tenets and as forming the basis of text A appears to contain a rich enough vocabulary to provide a platform for the convergence of views.

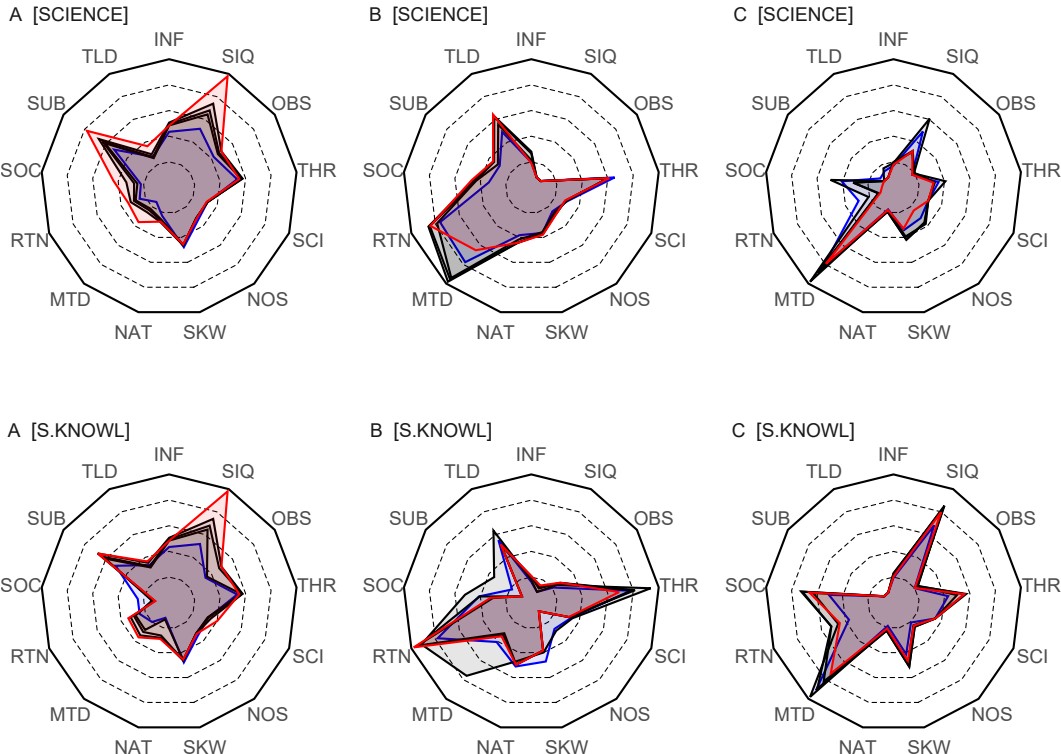

**Figure 7.** The lexicon profiles for T= [SCIENCE] and T = [SCI. KNOWLEDGE] in texts A, B, and C . The symbols for the properties P ∈ {F, P, D, e, Q, m, S, N, X} and how they are related to key words/terms is as in Tables 4, 6, and 8. The results are shown for local ($\beta = 0.2$, blue line) and for global connections ($\beta = 2.5$, red line), while the intermediate cases with $\beta = 0.5, 1.0$ and $1.5$ are shown in black lines.

## 4. Discussion and Conclusions

The many surrounding NOS are related to different understandings of the objectivity versus subjectivity of scientific knowledge, as well its certainty versus tentativity. In addition, the nature of scientific knowledge as constructed, and thus changing versus as approaching a durable, truth-like description of reality, is one of the central issues touched on in such debates (see, e.g., Reference [11,16,19,22–24]). Another topic included in the discussion about NOS is the relation of theories to laws and models [16,22,23]. While some of the discrepancies of views are based on fundamental metaphysical views on the nature of knowledge (see, e.g., Reference [16] for analysis), a substantial part of the disagreement may well lie in a lack of sufficient overlap between the words and terms used in discussions, and how the meaning of the words is understood; for example, what it means to be objective, subjective, tentative, or constructed. The ambiguity and context-dependence of such expressions also make the terms [SCIENCE] and [SCIENTIFIC KNOWLEDGE] polysemous and open up many ways to understand them. Such polysemy and the difficulty of establishing clear-cut definitions is obviously one reason behind difficulties in agreeing about NOS. Therefore, we suggest that it is useful to try to step away for a while from discussion that is heavily loaded with implicit background assumptions about science and scientific knowledge and simply turn to look at the skeletal lexical and syntactic structure of vocabularies used by the different parties involved in discussions about NOS. As we have argued here, we have good reason to assume that the basic vocabulary, in the form of a lexicon, can reveal much of the different stances and differences and similarities between different disciplinary schools [28–30].

The analysis of lexical networks and lexicon profiles is here used to reveal how texts by different authors about NOS for science education may use very different vocabularies when describing "science"

and "scientific knowledge". In extracting the vocabularies and lexicons for terms "science" and "scientific knowledge", we used network-based methods to analyze how the terms are connected to other words within the lexicon and thus how the other words support the meaning of the terms of interest. The analysis carried out is sensitive to the syntactical structure of the text, from the level of sentences up to that of context, where the syntactic level meets the semantic level. The connectivity of words and terms is here analyzed using network analysis based on (modified) communicability, which is used as a measure for lexical distance and for lexical importance. The viability of such a method based on communicability has recently been demonstrated in analyzing students' conceptual networks [31,37,38] and the conceptual content of students' study reports [32], as well as experts' didactical texts [33].

The network analysis presented in this study finds the global semantic connections between the terms and words, as well as groups them in thematic communities according to their semantic connectivity. Such analysis is neutral in the sense that groupings which emerge are purely based on semantic distances and not based on human evaluations of the meaning of the words. Although human evaluations of the meaning of words and terms are also based on semantic structures, the criteria on what basis the evaluations are done are not equally transparent and analyzable as in semi-automated and formal analyses based on network methods. The present analysis differs from previous ways to use semantic, network based methods in certain important aspects that deserve attention. In many analyses of semantic and lexical networks, the closeness of words is measured by the so-called closeness centrality, and, in such cases, lexical distance is simply taken to be the distance of co-occurrence of words [41–44]. Such analysis, however, is insensitive to the syntactical structure of the text, how cotexts are formed by sentences and contexts by the cotexts. The present choice to use a construct as shown in Table 2 attempts to retain the stratified structure of the texts. In that case, the counting of walks (or paths) in such a stratified structure is a more transparent way to measure the lexical distance than the closeness centrality [32,33]. The NOS conceptions of students and teachers have been examined by using network analysis previously providing results that show how their conceptions group around the basic tenets of NOS [43–45]. In those analyses, however, networks were used rather as visualizations and for example, the community structures were found just by qualitative visual inspection. While such approaches provide plausible results, are adequate for practical purposes, and their reliability can be asserted through inter-rater agreement, the underlying criteria of how connections and communities are detected lack transparency and analytical clarity the quantitative methods have. Moreover, the quantitative methods are more fine-grained and allow to find the community structure in finer details in comparison to qualitative, visual inspections. A more general discussion of advantages of quantitative network methods is beyond the scope of the present study, but such discussions have recently been reported elsewhere (see Reference [46]).

The quantitative network analysis of lexicons has of course its limitations. Such an approach leaves many issues untouched, most importantly how, for example, the choice of viewpoints of the authors of the texts affects the thematic organization of the terms and words and, consequently, how the meaning and polysemy depend on such choices. The analysis, while capable in revealing differences, remains silent of origins of those differences. On the other hand, this neutrality is an advantage of the method; it takes distance from views and debates rooted in preferred epistemological positions and implicit assumptions about the nature of knowledge. In that, it may help to resolve the unfruitful debates originating from the different ways to use terms "science" and "scientific knowledge" and to move towards more fruitful discussions, which focus on background reasons to construct the meaning of those words differently.

The results of the study show that the lexicons of the three texts A, B, and C are indeed very different in how they emphasize different lexical dimensions. In principle, the texts and their lexicons could complement each other, together providing a more complete picture of NOS. The lexicons are, however, so different that it is also possible that the texts represent such different views that they are difficult to merge; the vocabularies show divergence rather than convergence. In addition, the stratified

analysis of lexical structure is able to reveal how the lexicons may become augmented when the deeper contextual levels, by inclusion of more remote connections between the terms, are included in the analysis. The lexicons can also shrink, if the same idiosyncratic expressions and views are repeated on a deeper level. On the basis of vocabulary and lexicons, the texts indeed show signs of different schools of thought and the limited overlap of lexicons may be a sign of a certain isolation of the views as expressed in texts A, B, and C. From the text, we think, such a critical, even antagonistic, tone can be discerned. In particular, text B at times is so critical towards consensus NOS that it borders on rejection of consensus NOS views. On the positive side, the consensus view of NOS appears to possess a rich enough vocabulary on all essential dimensions such that building on it might provide a constructive and consolidated understanding of basic tenets of NOS, and through this incorporate the alternative views as proposed in texts B and C. One advantage of the lexicon provided by consensus NOS as it is organized around the seven tenets is that it guides a coherent organization of thematic and context-dependent use of terms and words: the thematic and context dependent community groupings parallel and are commensurate to a large degree. Such coherence is clearly an advantage for the practical purposes of education. Interestingly, a recent study utilizing network methods to analyze students' conceptions of NOS has found that the seven tenets of consensus can be identified through cluster (or community) analysis from the students' responses [43–45].

In summary, regarding the teaching of NOS and its tenets, the analysis of the lexical structures is a starting point to better understand how lexicons affect what kinds of conceptions are conveyed in teaching and instruction, and how lexicons may either facilitate or hinder discussions of certain aspects of NOS. The texts A, B, and C discussed here clearly differ in how richly they cover vocabulary to discuss different aspects of NOS; therefore, they also represent different emphases on that topic. Although this study is not meant to provide practical suggestions how to teach NOS, the main message is nevertheless that consensus NOS provides a scaffolding, where semantic meaning (understood as semantic fields) of terms related to "science" and "scientific knowledge" aligns well with their thematic clustering (communities). Such a concordance of key terms and thematic ordering are obviously desired features of well planned teaching strategies and designs. Taking the notion of importance of vocabularies and lexicons at face value, this suggests that different schools regarding NOS really exist, and, at the extremes, like those indicated by texts A and B, the lexicons are so different that constructive discussions and seeking consensus may prove very difficult. The results provided here suggest to us that the consensus view on NOS provides a good basis for consolidation of NOS and NOSK views, and it seems advisable to build on the tenets contained in consensus NOS, augmenting the vocabulary instead of replacing it with alternative though closely-related notions with diverging vocabularies. The analysis, which is neutral in respect of different epistemologies, may be warranted in this kind of situation and may represent a useful starting point for conceptual analysis, first on the level of vocabulary and lexicon, and then continuing to a deeper level of understanding of the basic notions.

**Funding:** This research received funding from Academy of Finland, grant 311449.

**Conflicts of Interest:** The author declares no conflict of interest.

## Appendix A. Communicability for Quantification of Lexical Distance

The communicability $G_{pq}$ between two nodes can be obtained from a $NxN$ adjacency matrix **A** with elements $[A]_{pq} = a_{pq}$, where $a_{pq} = 1$ when nodes are connected and $a_{pq} = 0$ when they are not connected. The powers $k$ of adjacency matrix **A** can be used to obtain the number of walks of length $k$ connecting two nodes within the network. In a connected network, however, the number of long walks increases rapidly, nearly factorially with the length of the walk. Therefore, the number of walks is usually divided by the factorial, to obtain the communicability [34,35,37,38]. For the walk counting, we use the the communicability matrix **G** with elements $G_{pq}$ between each pair of nodes $p$ and $q$. The communicability describes roughly how (e.g., information) content of node $p$ flows to

node $q$ [34,35,37,38]. Here, we use slightly modified communicability **G** where walks returning to nodes (self-reference) are removed, by defining the modified communicability as (cf. [32,33])

$$\mathbf{G}(\beta) = \sum_{k=1}^{\infty} \frac{\beta^k (\mathbf{A} - \mathbf{D})^k}{k!} = \mathrm{Exp}[\beta(\mathbf{A} - \mathbf{D})], \tag{A1}$$

where **D** is the diagonal matrix describing the number of connections attached to a given node (i.e., the node degree). The communicability centrality $G_p$ of a given node is obtained by summing the communicabilities to all other nodes, $G_p = \sum_q G_{pq}$. The parameter $\beta$ is used to tune how extensive a part of the network is included in counting the walks. By varying the parameter $\beta$, we obtain information on how the meaning attached to terms changes when only sentence-level connections are included ($\beta$ low) in comparison to the case when the context level is taken into account ($\beta$ high). It should be noted that, due to the finite size of the networks, after a certain sufficiently high value of $\beta$, the contribution from context-level does not change any more (compare with [32,33]). An optimum value of parameter $\beta$ is obtained when, with increasing value of $\beta$, one obtains the maximal growth rate of communicability. This value of $\beta$ is obtained when the Frechet-derivative [34,35] of **G** attains the largest value, usually occurring now with values $1.0 < \beta < 2.0$. In what follows, the communicability is normalized to a maximum value of one. By using the (normalized) communicability $[\mathbf{G}]_{pq} = G_{pq}$ between nodes $p$ and $q$, we can now obtain the total lexical support of node $q$ from all other nodes $p$, which are taken to be relevant in providing its lexical meaning.

The communicability of nodes within the lexical network is next used as the basis to form the lexical proximity network. In the lexical proximity network, we retain only those connections between nodes $p$ and $q$ that exceed a certain threshold $G^*$ of communicability. The lexical proximity network thus contains the most important lexical connections.

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
