# Peer review of "Usage of Terms “Science” and “Scientific Knowledge” in Nature of Science (NOS): Do Their Lexicons in Different Accounts Indicate Shared Conceptions?"

_education, doi:10.3390/educsci10090252_

Round 1

Reviewer 1 Report

The authors use a network approach to investigate consensus and differences on the concept of Nature of Science used in three text corpora. They justify the network approach by having a more objective perspective compared to close-reading and qualitative analyses as well as by Thomas Kuhn’s conception of lexical networks.

Major points

In some parts, the text remains inconcrete. For example, in the introduction the seven tenets of NOS should be mentioned and explained, especially, as criticisms against them are mentioned. They are mentioned later, but this explanation can be moved upwards.

The introduction remains rather inconcrete until the part of Thomas Kuhn’s lexicons, which are again clearly described. Thus I would suggest to add more examples and remove unnecessary philosophical terms from the introduction.

Line 76-81 should be removed from the introduction. The sentences appear exactly in the abstract, where they fit, but not in the introduction. Here, only a glimpse or outlook of the rest of the article would be appropriate.

The research questions can be tackled with this data and the selection of the texts used for analysis is described in a comprehensive way. However, it is not clearly enough described, why corpora C consists of three different texts, and how this affects the results.

I admit that I am not enough familiar with text based networks to judge the appropriateness of the applied methods. Nevertheless, the methods still entail a lot of subjective decisions, such as, which text passages to include, which centrality index to calculate, which thresholds to use or which property classes to assume. Such subjective decisions are inevident as in all scientific methods and by no means unwanted, as long as we are aware of this subjectivity. Thus, the mentioned advantage of more objectivity of the lexical network analyses is only partly an advantage.

Furthermore, it remains open, how other measures would have behaved in this context. Thus, we do not know, how robust the results are.

The results are described in a very detailed way for every text source, however, the explanations remain inconcrete and complicated except for the comparison of the text corpora. This paragraph, which is the most interesting part of the article, is clearly written. I would wish to see more such concrete and exemplified explanations throughout the other parts of the results section.

The article should discuss difficulties of the applied method more thoroughly. Also practical implications fall a bit short.

Minor points

Table 1 should include the year of the reference.

Line 87-89: these statements lack references

Figure 1: some colors such as light vs. dark purple are hardly distinguishable

Research Question 2 lacks auxiliary verbs: How DO the lexicons of A, B and C differ and where DO they agree? Both research questions lack question marks, although they are formulated as questions.

Figure 7: the “inner polygon” and “outermost polygon” is not so clear; the figure note should again state the colors to refer to values of beta.

Typos

l.22: are a topic

l.27: others

l.29: the nature of; organizes experience

l.30: the real

l.36: has

l.39: views

l.147: significantly, and

  1. 159: [one spacetab too much after full stop.]

l.176: the level of sentence up to the level

l.179: 2500 words [doubled word]

l.182: could “cotexts” (easily mixed up with contexts) be replaced by something like “sections”?

l.191: the terms of interest (T)

l.207: a set of nouns

l.208: from a given noun

l.209: [grammatically unclear; do you mean reverse linked?]

l.226: having more connections means

l.274: (e.g. models and laws).

l.279: [full stops missing after property classes]

l.288: that the division

l.289: while the division

l.294: [one spacetab too much after full stop.]

l.302: the more important it is in

l.308: they refer [doubled word]

l.331: refer to

l.346: Table 4 lists the words that are…

l.351: [one spacetab too much after full stop.]

l.361: into account

l.365: (i.e. how

l.366: thematic dimensions

l.379: that text A

l.397: in comparison to A.

ll.411: present in the lexical network

l.427: as it has already been visible from

l.433: to the whole text,

l.454: [one spacetab too much after full stop.]

l.467: 40%

l.471: thus are omitted in Table 7

l.516: [one spacetab too much after full stop.]

l.534: to [SCI. KNOWLEDGE],

l.544: to the same degree

l.563: vocabulary

l.569: on in such

l.570: in the discussion about

l.572: [one spacetab too much after full stop.]

l.612: [one spacetab too much after full stop.]

l.643: a NxN adjacency matrix

l.656: [one spacetab too much after full stop.]

l.660: (compare with[30,33]).

Table 6 contains some typos (especially 12. MTD)

The text contains typos, careless errors and grammatical inconsistencies. I highly suggest, to let an English native speaker read over the text again. Please check more carefully for typos next time BEFORE submission!

Further remarks

I want to point out, that such methods of text analyses often require much more time (and mathematical effort) than a qualitative analysis and close reading of the original sources; thus, the scientific value of such abstract analyses is scarce. Reading the article of about 14000 words takes more time than reading the analysed sources, which at the same time contain more information about the topic of NOS. If the analyses would comprise a lot more texts, the value of such an abstract analysis would increase again.

Therefore, I see methodological value in this article but less theoretical value/implications for educational science.

Regarding the findings on these texts, I would be very interested in what the original authors of the texts would say about these findings, and whether they would agree to what the network analyses revealed. But this is outside the scope of this article.

I hope my comments contribute to improve this work.

Best regards

Author Response

Thank you for reading and commenting the manuscript. I have attempted to respond as well as possible to all your comments and suggestions. I attach a pdf of responses that also include responses to other reviewers, because there are similar and useful points in reviewers questions and it may be helpful to see my responses to other reviewers. 

Reviewer 2 Report

Dear Author(s),

I would like to share some minor comments in the document. Also, you may share more concrete recommendations related to conclusion for the quality of paper.

Best regards...

Author Response

(The authors gave the same response as above.)

Reviewer 3 Report

I found this a very interesting approach for considering the different approaches to the Nature of Science (NOS), a well established research tradition that considers scientific knowledge, epistemology and describing the work of scientists.

The background section summarised the literature quite elegantly, and provided justification of taking a new approach to consider the problem of differing perspectives on the Nature of Science; what's important, what should be emphasised, and how.

The conclusions were interesting in that they explicitly compare different discussions of NOS from a linguistic point of view, finding where there was agreement and where linguistic use varied. The authors identify this variance as different fields or perspectives of NOS. This is important to identify because it helps makes explicit the different lens through which different researchers are characterising NOS. This in turn has consequences for practice, where NOS views are taught, practiced and assessed. Precision, therefore, around the theorisation and use of terminology, is important.

The main and only concern I had was really to do with the approach. I understand the basics of Network Analysis and can see that using this approach might be beneficial in viewing problems from a different perspective, but I felt as though the authors could do more to justify it's approach here. I acknowledge that this paper will be useful, in a methodological sense, to those using similar approaches, however, to contribute to the field of NOS, I felt that a little more had to be explicated.

For instance, in general, what does this approach provide that others don't? To me, the detailed analysis provided by the frequency and distance measures were very much dependent on the decisions the authors made in the selection of the texts and the interpretation of the terms identified. Corpus C, was said to include a great deal of variability, but it was also the only corpus that contained a number of texts from different authors, which explained a lot of this. Corpus B was similarly structured to Corpus A, though the authors state that this was because Corpus B was a refutation of A, so this made sense. Further, Corpus B showed significant disagreement with A, but the authors state that these disagreement 'is indeed evident from the general tone of text B'. I did wonder therefore why this approach was superior to any other linguistic analysis given that the salient linguistic features could very well have been just as well explicated in a number of ways (including simply comparing readings). What could the systems analysis do that these other approaches  can't?

Having not been able to read the author's own papers (referenced 29,30) hindered my ability to see how the analysis work in the examples of student writing, so this perhaps added to my confusion.

My suggestion would be to try and strengthen the justification of this approach, making it clear to the reader what benefit it gives us in understanding the underlying problem.

Author Response

(The authors gave the same response as above.)

Round 2

Reviewer 1 Report

Thank you for your detailed replies to my comments. You integrated many of my remarks, and if not, you reasoned well why you decided to remain with the original structure. Thanks as well for highlighting the most significant changes in the pdf.

The method is justified better and in general, the paper has a better flow and is aligned better.

Minor remarks

(in the address block): is ismo.Author@helsinki.fi really a valid e-mail address?

l.36: add a space in «1) the…» and remove a space before [31,32].

ll.37: I would use a lowercase “The” in the list throughout → 4) the…, 5) the… etc.

l.40: «form» or «are forming»

l.45: closing parentheses after [16,22] missing

l.51: recognizes

ll.62: they [doubled word]

l.80: from the viewpoint of lexicons

l.81: to pay attention

l.84: [26-28]. [full stop missing]

l.90: “but with regard to the…” or “but regarding the…”

l.128: in the late 1908s

ll.162: revise sentence structure

l.240: [remove fullstop after bracket]

ll.345: this indicates a multifaceted

l.371: [one spacetab too much after full stop]

l.380: theory [lowercase]

l.433: [SCIENTIFIC KNOWLEDGE]

l.490: the property class SOC

l.492: thus are omitted in Table 7,

l.527: relatedness. [full stop missing]

l.547: and by measuring

l.590: …-22]).

l.591: topic

l.618: terms and words, and groups them [the comma helps to read the text in the right tone with a pause after words]

l.621: are also based

l.632: network analysis

l.640: quantitative network methods

l.642: quantitative network analysis of lexicons has of course

ll.643: the choice of viewpoints of the authors of the texts affects

l.645: depend on such choices

ll.649: different ways to use the terms “science” and “scientific knowledge” and to move towards more fruitful discussions…

l.670: a large degree.

ll.789:

Koponen, I. T.; Nousiainen, M. Lexical Networks and Lexicon Profiles in Didactical Texts for Science Education. Complex Networks & Their Applications VIII: Proceedings of Complex Networks 2019, SCI 882. Cherifi, H., Gaito, S., Mendes, J. F., Moro, E. Rocha, L. M., Eds.; Springer International Publishing AG: Cham, Switzerland, 2019, pp. 15-27.

Check again for double spaces (especially after full stops like “Fig.”). In general, there is no need to abbreviate the short word “figure”.

Not a prerequisite but another idea:

The visualizations are quite complex, and require the reader to invest a lot of active work (e.g. mapping the abbrevations to the tables, going back and forth.

It would be really nice, if supplementary online material could reduce this effort, e.g. by a network that shows the full word and statistics when hovering over it. I am not sure whether this is possible with Mathematica as it requires some html and javascript code. As I said, this is out of scope of the other remarks but would be nice to have, maybe for future articles.

I am glad that my comments contributed to improve this work.

Best regards

Author Response

Thank you for final remarks and notions. I have now corrected the remaining typos pointed out by the reviewer (and the e-mail address) as suggested. Once more sincere thanks for such astonishing accuracy and thoroughness in reading the manuscript. I hope that now the most typos are eventually corrected.

(The abbreviations Fig. I have left as they are, because this is, however, very common).

The problem with spaces that appear additional seems to persist, but this seems to be uncorrectable. The problem probably is due to the way the mdpi.cls renders the pdf. In original LaTeX files there are no additional spaces in positions pointed out by the reviewer.   

The suggestion that dynamic pictures could be useful is certainly true. However, at present I have not found a way to extract such pictures from Mathematica. That would most probably need some additional coding. I keep this in mind for possible further manuscripts and try to find a way to do it.